# Effective resistance against pandemics: Mobility network sparsification for high-fidelity epidemic simulations

**Alexander Mercier** [1,2]*, **Samuel Scarpino**[2,3,4,5,6], **Cristopher Moore**[2]

**1** Department of Mathematics & Statistics, University of South Florida, Tampa, Florida, United States of America, **2** Santa Fe Institute, Santa Fe, New Mexico, United States of America, **3** Pandemic Prevention Institute, The Rockefeller Foundation, Washington, D.C., United States of America, **4** Network Science Institute, Northeastern University, Boston, Massachusetts, United States of America, **5** Department of Physics, Northeastern University, Boston, Massachusetts, United States of America, **6** Vermont Complex Systems Center, University of Vermont, Burlington, Vermont, United States of America

* amercier1@usf.edu

## Abstract

Network science has increasingly become central to the field of epidemiology and our ability to respond to infectious disease threats. However, many networks derived from modern datasets are not just large, but dense, with a high ratio of edges to nodes. This includes human mobility networks where most locations have a large number of links to many other locations. Simulating large-scale epidemics requires substantial computational resources and in many cases is practically infeasible. One way to reduce the computational cost of simulating epidemics on these networks is *sparsification*, where a representative subset of edges is selected based on some measure of their importance. We test several sparsification strategies, ranging from naive thresholding to random sampling of edges, on mobility data from the U.S. Following recent work in computer science, we find that the most accurate approach uses the *effective resistances* of edges, which prioritizes edges that are the only efficient way to travel between their endpoints. The resulting sparse network preserves many aspects of the behavior of an SIR model, including both global quantities, like the epidemic size, and local details of stochastic events, including the probability each node becomes infected and its distribution of arrival times. This holds even when the sparse network preserves fewer than 10% of the edges of the original network. In addition to its practical utility, this method helps illuminate which links of a weighted, undirected network are most important to disease spread.

## Author summary

Epidemiologists increasingly use social networks to understand how geography, demographics, and human mobility affect disease spread and the effectiveness of intervention strategies. While highly detailed data on human social networks are now available, the size and density of these modern networks makes them computationally intensive to study. To address this challenge, we study methods for reducing a network to its most important

**Data Availability Statement:** All data and code used for running network sparsification, models, and plotting is available on GitHub at https://github.com/AMMercier/EffectiveResistanceSampling.

**Funding:** This study was funded by the National Science Foundation via two grants (https://nsf.gov/index.jsp, https://nsf.gov/index.jsp) 1757923 and 1838251 to CM. The sponsors or funders have played no role in the study design, data collection and analysis, decision to publish, or preparation of the manuscript.

**Competing interests:** The authors have declared that no competing interests exist.

links. Following recent work in computer science, we use the effective resistance, which takes both local and global connectivity into account. We test this method in simulations on a U.S.-wide mobility network and find that it preserves epidemic dynamics with high fidelity. Combined with efficient epidemic simulation algorithms, our approach can facilitate a more effective response to epidemics.

This is a *PLOS Computational Biology* Methods paper.

## Introduction

Networks are a powerful tool for understanding the effects of superspreading events, geographic and demographic communities, and other inhomogeneities in social structure on the spread of infectious diseases [1–4]. As a consequence, network-based models for simulating epidemics have become particularly popular. However, simulating a stochastic epidemic model typically takes time proportional to the number of edges or links along which the disease might spread [5]. This makes these models computationally expensive for dense networks where most nodes have edges of nonzero weight to many destinations, such as those derived from high-resolution mobility data [6,7]. This computational cost is exacerbated by the need to perform many independent runs on the same network to get a sense of the probability distribution of events–for instance to calculate the probability that each individual becomes infected or to test the effect of various intervention strategies and different initial conditions [8].

A natural way to reduce this computational cost is *sparsification*: choosing a subset of important links to produce a sparse network whose behavior is faithful to the original, but which is less costly to study. One popular method is simply to remove links whose weights are below a certain threshold (see [9] for an overview). This is intended to remove low-weight links that are unlikely to spread the contagion, or which have nonzero weight simply due to noise in the measurement process. However, it is unclear to what extent this naive thresholding approach preserves the true behavior of contagion spread. In particular, thresholding ignores the "strength of weak ties" [10]: a low-weight edge could play a important role in low-probability, but high-impact, events if it is one of the few ways to spread the epidemic from one region or community to another.

A more sophisticated family of sparsification algorithms comes from recent results in computer science. These algorithms efficiently solve large systems of linear equations by making them sparser, i.e., by choosing and reweighting a random subset of their terms or coefficients. By sampling in a specific way, the spectrum of the linear system can largely be preserved and thus approximate the solution with high accuracy [11,12].

In the context of networks, we can think of these algorithms as follows. Given a weighted, undirected network $G$ with $n$ nodes and $m$ edges, we assign each edge a probability $p_e$ of being included in the sparsified network. This probability might depend on the entire network, not just on the weight of that edge, which we denote $w_e$. Let $q$ be the fraction of edges of $G$ we wish to preserve. Then, we form a sparse network $\tilde{G}$ on the same set of nodes by sampling $s = qm$ edges independently from the distribution $\{p_e\}$. If an edge $e$ is chosen, we set its weight in $\tilde{G}$ to $\tilde{w}_e = w_e/(p_e s)$. An edge may be chosen multiple times, in which case we sum $\tilde{w}_e$ accordingly (so the number $\tilde{m}$ of distinct edges in $\tilde{G}$ is less than or equal to $s$). This reweighting from $w_e$ to $\tilde{w}_e$ compensates for the fact that the network is sparser overall, while ensuring that $\tilde{w}_e$ equals

the original weight $w_e$ in expectation. Similarly, the weighted adjacency matrix and graph Laplacian of $\tilde{G}$ are equal, in expectation, to those of $G$.

It might seem strange to use random choices, rather than a deterministic criterion, to decide which edges of the original network to include in the sparsification. But the hope is that even if the fraction $q$ of preserved edges is much smaller than 1, these choices cause the epidemic behavior of the network to concentrate around that of the original network, rather like the Central Limit Theorem makes empirical means converge after a moderate number of samples.

The question remains how to assign the probabilities $p_e$. One choice is to make $p_e$ proportional to the weight $w_e$. An even simpler choice is to make these probabilities uniform, $p_e = 1/m$ for every $e$. However, neither of these choices takes into account whether $e$ is structurally important—for instance, if it is the only way to cross between two communities or is redundant, with many alternate paths that play the same role.

Graph theorists and network theorists have invented various kinds of "betweenness" to measure the structural importance of an edge (see [13] for a review). However, edge betweenness takes $O(nm)$ time to calculate for graphs with $n$ nodes and $m$ edges [14], making it impractical for large, dense networks. There is also a rich literature on identifying a "backbone" or "effective graph" of a network, in order to summarize its structure, preserve its statistical properties, or identify causal connections (e.g., [15–17]).

In particular, the distance backbone defined in [16] preserves all shortest-path distances on a weighted graph. This is clearly important to many types of dynamical systems on a network. On the other hand, epidemic spread is a setting in which many parallel paths can combine to transmit a disease more quickly than a single shortest path. Our goal is to sparsify networks in a way that takes this effect into account.

Here we consider a sparsification algorithm with rigorous guarantees for spectral properties, and therefore for linear dynamics, due to Spielman and Srivastava [11] (simplifying earlier work by Spielman and Teng [12]). It uses the edges' *effective resistance*, denoted $R_e$. Effective resistance can be understood by transforming the given network into an electrical circuit where each edge $e$ becomes a resistor with resistance equal to $1/w_e$. Then $R_e$ is the resistance of this network between $e$'s endpoints. This takes into account not just $e$, but all other possible paths between the endpoints of $e$, each of which reduces $R_e$ as in a parallel circuit. If $e$ is the only path between its endpoints, then $R_e = 1/w_e$. If there are many alternate paths that are short and consist of high-weight edges, then $R_e$ is small.

The effective resistance $R_e$ and the product $w_e R_e$ have many names in different fields, e.g., [18–22], including information distance, resistance distance, statistical leverage, current flow betweenness, and spanning edge betweenness; this last because, due to Kirchhoff's matrix-tree theorem, $w_e R_e$ is the probability that a random spanning tree includes $e$ if each spanning tree appears with probability proportional to the product of its edge weights [23]. Moreover, $R_e$ can be computed for all edges simultaneously by inverting the graph Laplacian and can be approximated in nearly-linear time using a random projection technique (see Methods). The Spielman–Srivastava algorithm chooses edges with probability $p_e$ proportional to $w_e R_e$. This prioritizes edges that are the only efficient way to travel between their endpoints, for which $w_e R_e \approx 1$. Intuitively, this strategy helps keep the network connected and preserves its global structure. Indeed, in [11] they proved that sampling just s = $O(n \log n)$ edges gives a Laplacian very close to that of the original network, making it possible to solve certain large systems of equations in nearly-linear time [12].

In our setting, if the original network is very dense with $m = O(n^2)$ edges then we can radically sparsify it, reducing its average degree from $O(n)$ to $O(\log n)$ and keeping just a fraction $q$

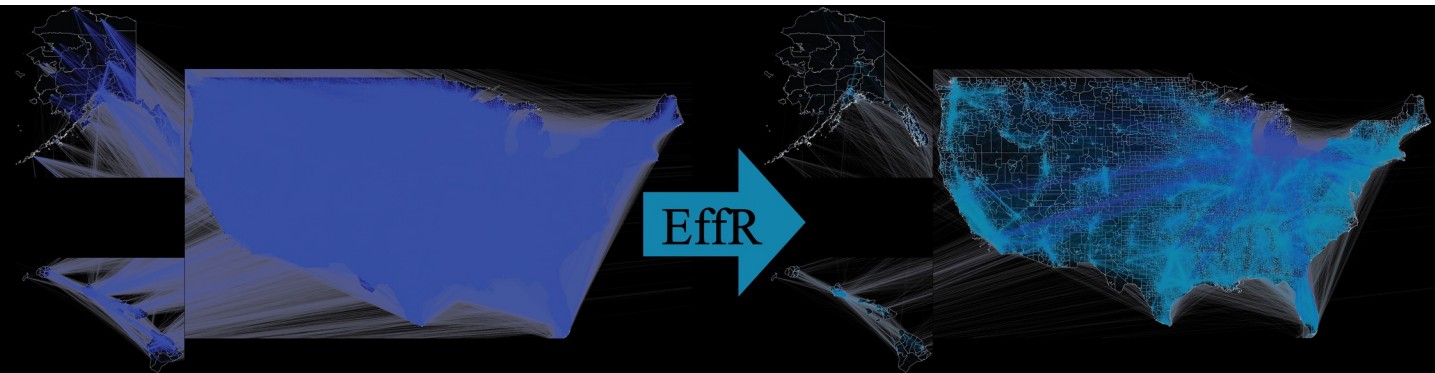

**Fig 1. Sparsification of the U.S. mobility network.** The U.S. mobility network, where nodes are census tracts and edges are weighted according to average human mobility between census tracts. On the left, the original network with about 26 million edges. On the right, a sparsified network based on effective resistance sampling with $q = 0.1$, preserving about 7% of the edges of the original. Heavier-weight edges are lighter in color on a logarithmic scale. Note the mixture of local and long-range links, and how sparsification reweights edges to be heavier in order to compensate for the decrease in density.

$= O((\log n)/n)$ of the edges, shown graphically in Fig 1. However, an ideal sparsifier will decrease the density of a network while preserving both structural and dynamical properties, so that models will behave similarly on the sparsified network as they do on the full network. Said differently, the utility of the Spielman–Srivastava algorithm for our problem requires that $w_e R_e$ be a reasonable measure of edge importance in an epidemic. It is well established in network epidemiology that the spectrum of the Laplacian or adjacency matrix can be used to determine epidemic thresholds (e.g., [24–26]) since this is a matter of linear stability. However, epidemic models such as SIR are nonlinear as well as stochastic. Therefore, when evaluating the performance of various sparsification algorithms, we focus on measures that provide a richer description of the epidemic dynamics.

Exploring whether effective resistance is a good measure of edge importance in the epidemic context is the goal of this paper. Some early work in this direction was done by [27] using Hamming distance as a metric of sparsifier performance. Here we confirm on a dense real-world mobility network that the Spielman–Srivastava algorithm preserves the behavior of the SIR model, even when we keep only a few percent of the original edges. This includes both the bulk behavior of the epidemic and the probability distribution of events, including the probability that each node becomes infected and the distribution of times at which it does so. According to these metrics, it achieves higher fidelity than simpler edge-sampling methods where the probability is uniform or proportional to edge weight, as well as the naive thresholding approach. While estimating the effective resistance for each edge has a higher computational cost than these methods, this cost is only incurred once for each network as a preprocessing step, after which independent trials of the SIR simulation (and, if desired, of the edge-sampling process) can be carried out at no additional cost.

In addition to reducing the computational effort required to carry out accurate simulations, these results offer some insight into which links are important for disease spread, both topologically and quantitatively.

## Materials and methods

### Commuter network data set

The real-world mobility network was constructed from publicly available United States Census Bureau inter-census-tract commuting flows for all fifty states. Each node is a single census tract, and integer edge weights denote the amount of inter-census-tract human mobility

provided by the United States Census Bureau through a summary of Longitudinal Employer-Household Dynamics (LEHD) Origin-Destination Employment Statistics (LODES) across Origin-Destination (OD), Residence Area Characteristic (RAC), and Workplace Area Characteristic (WAC) data types for the year 2016. This commuting data is directional, so *a priori* this network is directed. Since effective resistance and many other sparsification techniques assume an undirected graph (which is standard in the field of network epidemiology [28]), we set the undirected edge weight for $(i, j)$ to the average of the directional weights between $i$ and $j$ in each direction. The resulting network is comprised of $n = 72,721$ nodes and $m = 26,319,308$ edges, giving it an average topological degree of $2m/n = 723.8$. Each node has three pieces of metadata: the population of the census tract, its land area in square meters, and its Geographic Identifier (GEOID). Standardized census tract GEOIDs are used to merge node data with United States Census Bureau cartographic boundary shape files of all fifty states and assign corresponding Rural-Urban Commuting Areas (RUCA) codes, standardized by the Economic Research Service of the United States Department of Agriculture. These codes range from 1 (urban core) to 10 (rural) and summarize the level of urbanization, daily commuting, and population density of the given census tract according to United States Census Bureau standards.

## SIR simulation algorithm

To simulate the SIR model on large, complex networks, we use a continuous-time, event-driven Gillespie algorithm [29]. This algorithm stores a "heap" of potential events; a heap is an efficient implementation of a priority queue that allows events to be added or removed in $O(\log n)$ time. At each step the event in the heap with the smallest (soonest) time becomes the current event. It is removed from the heap, and new events driven by that event are added to the heap. Specifically, whenever a node becomes infected, infection events for all its neighbors are added to the queue along with its own recovery event. If an infection event $i \rightarrow j$ is drawn from the heap, we check that $i$ is still infected and $j$ is still susceptible; this is simpler than removing these events from the heap when, say, $i$ recovers.

If the overall infection rate is $\beta$, each edge $e$ transmits at a rate $\beta w_e$, and we assume all nodes have the same recovery rate $\gamma$. New events are given a time equal to the current time $t$ plus a waiting time $\Delta t$, where $\Delta t$ is drawn from an exponential distribution with mean $1/(\beta w_e)$ or $1/\gamma$ for infection and recovery events respectively. This exponential distribution assumes that infection and recovery are continuous-time Markov processes with constant rate, but the same approach easily generalizes to more complicated waiting time distributions.

For analysis, each run of the SIR simulation outputs whether each node becomes infected, and its arrival time if so. It also outputs the total number of nodes susceptible, infected, or recovered as a function of time, the total computation time on the CPU, and the maximum heap size.

## Network sparsification

We use two types of sparsification methods: weight thresholding and edge sampling. Both methods focus on edge reduction and do not change the number of nodes. Weight thresholding removes all edges below a specified weight. For comparison with the edge sampling sparsifiers, we vary this threshold to preserve a given fraction of edges.

We employ three edge sampling algorithms: uniform sampling, sampling by weight, and sampling according to effective resistances. These sampling algorithms utilize the same general scheme, but vary in the amount of information they consider about the given edge, ranging from most naive (uniform) to the most sophisticated (effective resistances). Each sampling procedure computes an importance $u_e$ for each edge, where $u_e$ for the three methods is shown in Table 1. It then samples $s$ edges with replacement with probability $p_e = u_e / \Sigma_{e'} u'$ and adds

**Table 1. Random Sampling Importances.**

| Sampling Procedure | Edge Importance $u_e$ |
|---|---|
| Uniform (Uni) | 1 |
| Weights (Wts) | $w_e$ |
| Effective Resistance (EffR) | $w_e R_e$ |

the sampled edges to the sparse network with weight equal to $\tilde{w}_e = w_e/(p_e s)$, shown in Algorithm 1.

---

Algorithm 1: Edge-Sampling Sparsification

---

**Input:** dense network $G(V, E, \phi)$
**Output:** sparse network $\tilde{G}(V, \tilde{E}, \tilde{\phi})$
**Parameters:** number of samples $s$, and edge importances $\{u_e\}$
**Procedure:**
Choose random edge $e$ from $G$ with probability $p_e \propto u_e$
Add edge $e$ to $\tilde{G}$ with weight $\tilde{w}_e = w_e/(p_e s)$ Take $s$ samples with replacement
Sum weights if an edge is chosen more than once

---

Since edges are chosen with replacement, the same edge may be sampled multiple times: in that case its weight is summed, i.e., multiplied by the number of times it is sampled.

Reweighting edges from $w_e$ to $\tilde{w}_e$ in this way ensures that $\langle \tilde{w}_e \rangle = w_e$, and therefore that $\langle \tilde{A} \rangle = A$ and $\langle \tilde{L} \rangle = L$ where these are the adjacency matrix and Laplacian of $\tilde{G}$ and $G$ respectively. Thus, the sparsifier preserves the linear properties of the original network in expectation. Moreover, Spielman and Srivastava [11] showed that if we sparsify using effective resistances and $s$ is a sufficiently large constant times $n \log n$, then $\tilde{L}$ is concentrated around $L$ in a powerful sense. Let $\varepsilon > 0$ be the desired error in the Laplacian. Then if $s \geq 8n \log(n/\varepsilon^2)$, then with probability at least 1/2 (and tending to 1 if $s$ increases further) for all vectors $x \in \mathbb{R}^n$ we have

$$(1 - \varepsilon)x^T L x \leq x^T \tilde{L} x \leq (1 + \varepsilon)x^T L x \tag{1}$$

As a result, $\tilde{L}$ approximately preserves the important eigenvectors and eigenvalues of the original Laplacian $L$.

## Effective resistance

The effective resistance between any two given nodes is the resistance across them in a network of resistors where each edge has conductance $w_e$ or equivalently resistance $1/w_e$. The entire matrix of effective resistances can be computed from the graph Laplacian as follows: the effective resistance for an edge $e = (i, j)$ is

$$R_e = (e_i - e_j)^T L^+ (e_i - e_j) \tag{2}$$

where $L^+$ is the pseudoinverse of $L$ and $e_i$ is the basis column vector with 1 at position $i$ and zeros elsewhere.

Computing the exact pseudoinverse of $L$ is computationally expensive for large networks. Therefore, Spielman and Srivastava [11] approximate the effective resistances using a random projection technique. This yields approximate resistances $R_e$ such that, for all edges $e$

$$(1 - \varepsilon)R_e \leq R'_e \leq (1 + \varepsilon)R_e \tag{3}$$

where $\varepsilon$ can be made as small as desired at increased computational cost. These approximate

resistances can then be used by the Spielman–Srivastava algorithm with a slightly larger error parameter $\varepsilon$ in (1).

We carried out this approximation for $\varepsilon = 0.1$ using the implementation of Koutis et al. [30], which takes time $\tilde{O}(m \log(1/\varepsilon))$ where $\tilde{O}$ hides factors logarithmic in $n$. This procedure took several hours for the U.S. commuting network: note that we only need to do this once, after which we can generate sparsified networks with any desired density and run the SIR model on each one as many times as we like.

## Experiments and parameters

To compare the four sparsification methods, ten sparse graphs of each type were created with a varying number of edges. For thresholding, we set the threshold in order to retain a given fraction of the edges, ranging 0.2 to 0.9 in steps of 0.1. Since these sparsifications performed poorly compared to the edge-sampling methods, we did not reduce the density further.

For the edge-sampling methods, we set the number of samples for each method to $s = qm$ where $q = 0.001, 0.00325, 0.0055, 0.00775, 0.01, 0.0325, 0.055, 0.1, 0.55$, and $1$. Since edges can be sampled more than once, the fraction of edges preserved by the resulting sparse network can be slightly smaller.

We chose two representative initial conditions, where the initial infections are localized or dispersed. The localized initial condition consists of a single infected node, namely the census tract containing John F. Kennedy International Airport. In the dispersed initial condition, we infect 1% of the nodes chosen with probability proportional to their population.

We chose representative values of the parameters $\beta$ and $\gamma$ in order to create a nontrivial distribution of infection probabilities and arrival times, thus posing a challenge for sparsification. In both cases we set the recovery rate to $\gamma = 1$. We set $\beta \approx 0.0014$ and $\beta \approx 0.0046$ for the localized and dispersed initial condition, respectively.

Since each edge with weight $w_e$ transmits the infection at rate $w_e \beta$ rate, and since the average weighted degree of a node in the network is $\bar{w} \approx 1782$ these correspond to reproductive numbers $R_0 = \bar{w}\beta/\gamma = 2.50$ and $R_0 = 8.20$ respectively, although of course in a heterogeneous network no single value of $R_0$ is sufficient to model the dynamics [31].

For each sparsified network and each initial condition, 1000 simulations were run for analysis. We ran the simulation for a maximum time $t_{\max} = 20$: in most but not all simulations, all nodes were either recovered or susceptible at that point.

## Sparsifier evaluation

For each sparsifier, the fidelity of the stochastic SIR dynamics to that on the original network was assessed through three metrics: the probability of infection for each node across 1000 simulations, the distribution of arrival times for each node, and the average SIR curve across all simulations.

We compared the infection probabilities with scatterplots as shown in Fig 2, where for each node the horizontal and vertical coordinates are its infection probability in the original and sparsified network respectively. If the sparsifier preserved these probabilities exactly, all nodes would fall on the diagonal. We measured various quantities such as the squared correlation $R^2$ and the $L_1$ and $L_2$ distances to confirm that these probabilities are approximately preserved. The arrival time distributions, i.e., the distributions of the time at which each node becomes infected, were also compiled 1000 simulations.

We compared these distributions using the *Arrival Time Error Score* (ATES) averaged over all nodes. As stated in the main text, for each node this score is the Wasserstein distance between the arrival time distributions conditioned on the node becoming infected, with a

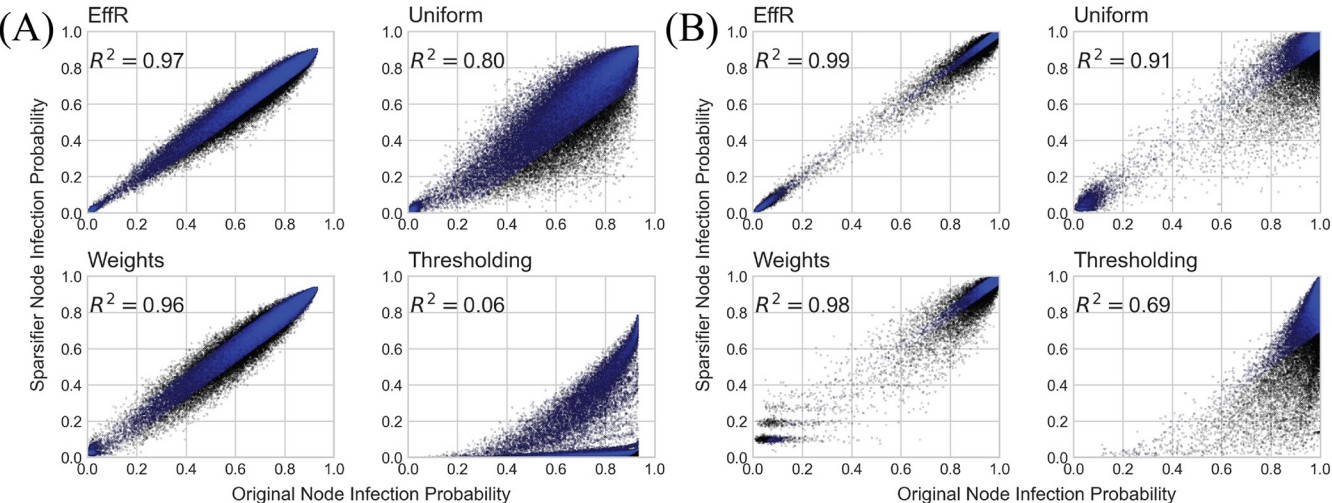

**Fig 2. Sparsification of the U.S. mobility network: preserving infection probabilities.** Scatterplots of infection probabilities for localized (A) and dispersed (B) initial conditions for four sparsification methods: effective resistance, uniform sampling, weight-based sampling, and simple thresholding. For each sampling method we chose $0.1m$ edges of the original network, and we set the threshold to retain the 10% highest-weighted edges. Each dot represents a node in the network. The horizontal and vertical axes give the probability that node becomes infected using the original and sparse network respectively, based on 1000 independent runs of the SIR model. Dots close to the diagonal are those for which sparsification preserves this probability, and the blue dots represent the 90% of nodes closest to the diagonal. While weight-based sampling achieves a similar $R^2$, it is not as good at preserving infection probabilities for low-probability nodes, especially for dispersed initial conditions. Effective resistance preserves the infection probability for both low- and high-probability nodes.

penalty $t_{\max}$ if the node has a zero infection probability in one network but a nonzero infection probability in the other, so that one of the two arrival time distributions is undefined. Typically, this occurred because the sparsifier cut that node off from the rest of the network. If a node has zero infection probability in both networks, we assign an error score of zero. Thus the ATES is small only if the sparsifier is faithful to the stochastic behavior of the SIR model on the original graph in two senses: 1) they agree on which nodes are infected with nonzero probability, and 2) when a node becomes infected, they agree on the distribution of times at which it does so.

The Wasserstein distance of two distributions is the average difference between a point from one distribution (in this case, an arrival time) and the corresponding point in the other distribution, minimized over all possible probability-preserving correspondences. This is also called the "earthmover distance" and comes from optimal transport theory [32]: it is the minimum, over all ways to transport earth from one pile to another, of the average distance we have to move each unit of earth. Formally, if $X$ and $Y$ are two random variables, their Wasserstein distance is

$$W(X, Y) = \inf_{\pi \in \Gamma(X,Y)} \int |x - y| d\pi(x, y) \tag{4}$$

where $\pi$ is minimized over all couplings between the two distributions, i.e., over the space $\Gamma(X, Y)$ of joint distributions on $X \times Y$ whose marginals over $X$ and $Y$ are their distributions.

For distributions of one real-valued variable such as the arrival time, the optimal coupling is simple: we match times in the two distributions according to their quantiles, i.e., their cumulative distribution functions $C(x) = Pr[X < x]$ and $D(y) = Pr[Y < y]$. Then

$$W(X, Y) = \int_0^1 |C^{-1}(z) - D^{-1}(z)| dz \tag{5}$$

Numerically, for each node we take the list of arrival times at which it became infected in the original network (from the subset of simulation runs in which it did so) and the analogous list in the sparsified network. We sort each list from smallest (earliest) to largest, giving times $s_1 < s_2 < \ldots$ and $t_1 < t_2 < \ldots$. If these lists are the same size $\ell$, then $W(X, Y)$ is just the average over all $1 \le i \le \ell$ of $|s_i - t_i|$. (If the lists are of different sizes $\ell_1$ and $\ell_2$, we rescale them to produce two empirical distributions, assigning $s_i$ in part to $t_{\lfloor (\ell_2/\ell_1)i \rfloor}$ and in part to $t_{\lceil (\ell_2/\ell_1)i \rceil}$.) Thus, for each node, $W(X, Y)$ is the absolute difference in its arrival time for the two networks, averaged over all simulations, conditioned on the event that it becomes infected.

We note that some other common measures of distance between probability distributions, such as the Kullback-Leibler divergence, do not pay attention to the magnitude of the temporal error. For instance, the KL divergence is large for two arrival time distributions that are both narrowly peaked regardless of whether the distance between those peaks is large or small. Similarly, the KL divergence is low between two distributions with a large overlap, regardless of how severe the temporal difference is in their non-overlapping parts.

## Results

We focus our attention on a U.S.-wide network of commuting patterns based on data from the U.S. Census Bureau (see Methods). Mobility data has proven a powerful tool for incorporating realistic social contact and population connectivity into epidemiological models [33–35], which has been especially true during the SARS-CoV-2 pandemic [36–41]. For example, [42] showed that reductions in commute flow correlated with lower SARS-CoV-2 prevalence in New York City. However, it is well established that commute flows alone are almost certainly too simplistic to capture the dynamics of modern epidemics in humans [3,43]. Thus, while we treat this network as a test case for sparsification in modern data drawn from human social structure because it is large, dense, and highly heterogeneous, we do not claim that it is an accurate representation of the epidemiological contact network. In particular, our simulations simplistically assign a single state (Susceptible, Infected, or Recovered) to each census tract. On the other hand, a more realistic model where each census tract has population-level variables (the fraction of people in that tract with each SIR state) would also be simplified by sparsification, since each edge corresponds to a coupling term in the resulting set of differential equations. This network has roughly 73 thousand nodes, each corresponding to one census tract, and roughly 26 million edges. Edge weights are given by commuting flows averaged over 2016. Its average topological degree (the number of connections with nonzero weight) is 724. Fig 1 shows a sparse version of this network, using effective resistance to sample $s = qm$ edges with $q = 0.1$. Since some edges are chosen multiple times, this process preserves about 7% of the original number of edges.

We simulated an SIR model both on this network and on sparse networks generated by the three edge-sampling methods described above, where probabilities are uniform, proportional to edge weight, or based on effective resistance. For comparison, we also included the simple threshold method. In each case, we varied the fraction of edges preserved by varying either the weight threshold or the fraction $q$ of edges sampled.

Our SIR simulations are continuous-time, event-driven, stochastic Markov processes. Each edge $e$ transmits the disease at rate $\beta w_e$, and each infected node recovers at rate $\gamma$. We consider two types of initial condition: a localized one, where only the node corresponding to JFK International Airport in New York City is infected, and dispersed one, where 1% of the nodes (a total of 727) are chosen with probability proportional to their population. We chose representative values of $\beta$ and $\gamma$ to create a wide range of probabilities with which individual nodes become infected, and a wide range of arrival times at which they do so.

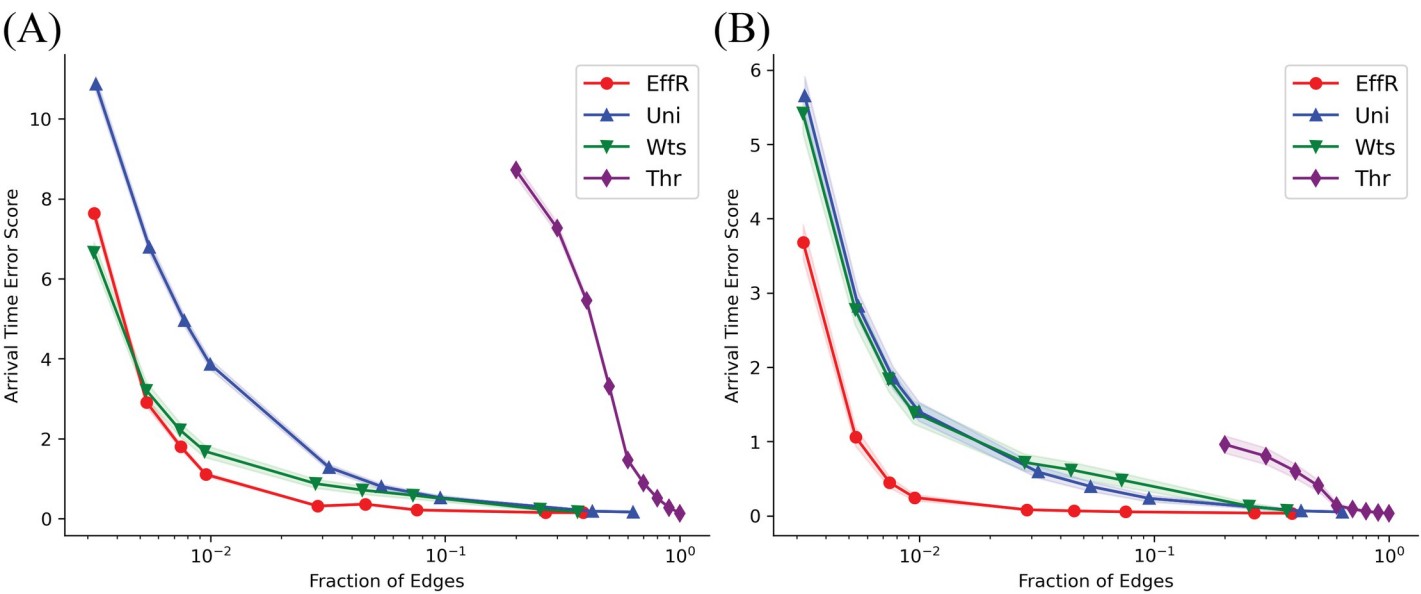

**Fig 3. Sparsification of the U.S. mobility network average: Arrival Time Error Score.** Arrival Time Error Score averaged over nodes and over 1000 independent simulations for the (A) localized initial conditions and (B) dispersed initial condition. The shaded regions correspond to one standard deviation of the average, which for several of these curves is too small to see. The horizontal axis shows the fraction of distinct edges of the original network preserved by the sparsifier. Note that this fraction is slightly less than $q$, since edge-sampling algorithms can choose the same edge multiple times.

Since epidemics are inherently stochastic, we are not only concerned with macroscopic quantities like the fraction of the population that is susceptible or infected at a given time. We also wish sparsification to preserve important aspects of the probability distribution of events. In particular, we are concerned with the probability that each node becomes infected and the distribution of arrival times when it becomes infected. In order to have a distribution of events for each node, simulations were run independently 1000 times for each set of initial conditions and for each sparsifier.

In Fig 2, we show how the three sampling methods and simple thresholding perform at the task of preserving infection probabilities. As in Fig 1, we preserve about 7% of the distinct edges of the original network. Each dot represents a node, with the probability it becomes infected on the original network and the sparsified network plotted on the horizontal and vertical axes, respectively. Dots on the diagonal are those for which these probabilities are the same. We see that sampling with effective resistances accurately preserves these probabilities across the entire distribution, from low to high probability, and in both initial conditions. Weight-based and uniform sampling perform reasonably well, but with larger error shown by the distance of these dots from the diagonal. Naïve thresholding performs quite poorly, even while containing 10% of the original edges.

We are also interested in the arrival time distribution of each node, i.e., the distribution of times at which it becomes infected [44,45]. We define the error with which a sparsification method preserves this distribution as follows. First, we use the Wasserstein distance to compare the arrival time distributions of each node in the original and sparse network, ignoring the runs of the SIR model in which it never becomes infected. The Wasserstein distance is essentially the average difference between the times at which the node becomes infected in the two networks (see Methods for a formal definition). However, if a node's infection probability is nonzero in one network but zero in the other so that its arrival time distribution is empty, we impose a penalty equal to the duration $t_{max}$ of the simulation (the maximum possible

Wasserstein distance). We call the average of this quantity across all nodes the Arrival Time Error Score. Its value is low if the same nodes have nonzero infection probability in both networks—in particular, if the sparsifier doesn't cut potentially infected nodes off from the rest of the network—and if each such node becomes infected at a similar distribution of times.

We show this error score, averaged over 1000 runs of the SIR model, for the four sparsification methods in Fig 3 for the localized (A) and dispersed (B) initial conditions. In both cases, even when we only preserve a few percent of the original edges, sampling by effective resistances (EffR) achieves a small error. Uniform and weight-based sampling also perform reasonably well on this measure. However, in the dispersed initial condition both are significantly worse than effective resistance when we preserve less than 10% of the original edges. In the localized initial condition, weight-based sampling is comparable to effective resistance, but uniform sampling is significantly worse below about 5%.

For illustration, in Fig 4 we show the arrival time distributions for two representative nodes. We compare these distributions on the original network with the three edge-sampling sparsifiers. All three reproduce the shape and width of these distributions fairly well, giving a small Wasserstein distance. However, since these distributions are conditioned on the event that these nodes become infected, this comparison does not measure whether the infection probabilities are the same.

Indeed, as shown in Fig 5, the main reason why uniform and weight-based sampling have a larger error is that they disconnect a significant fraction of the nodes from the rest of the network, giving them a zero infection probability.

In contrast, effective resistance keeps almost all nodes connected even when preserving just 1% of the original edges.

We also studied the performance on these sparsifiers on census tracts of different types. We use the RUCA (Rural-Urban Commuting Area) codes of the U.S. Department of Agriculture,

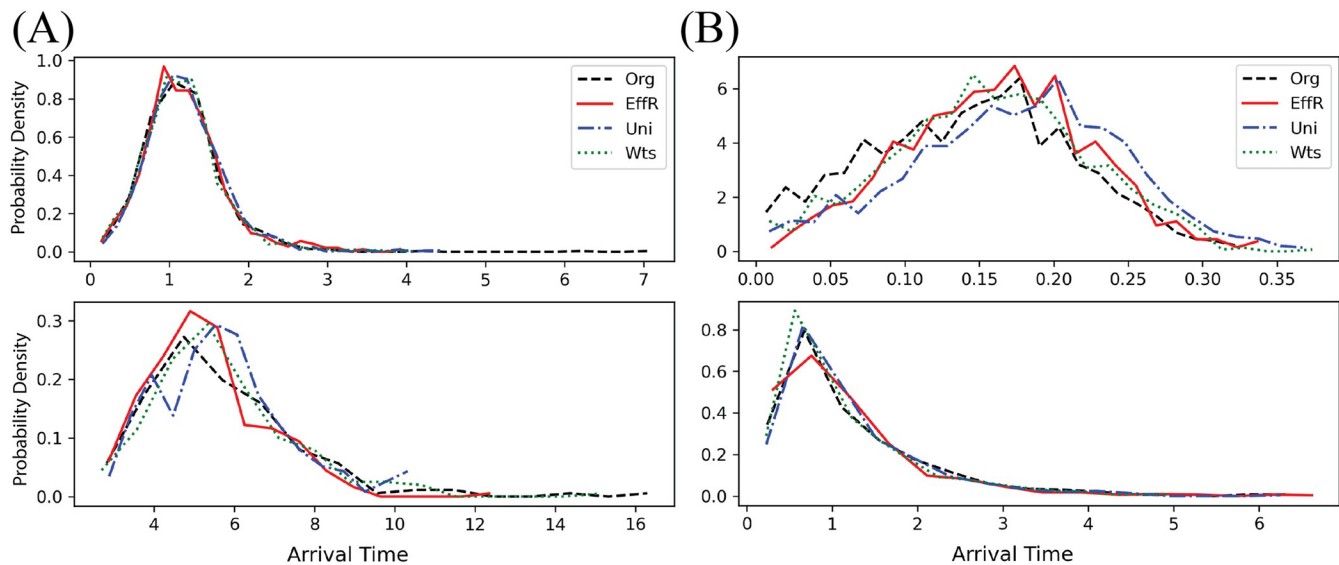

**Fig 4. Sparsification of the U.S. mobility network: representative arrival time distributions.** Arrival time distributions for the original network (Org) and sparsified networks produced by the three edge-sampling methods with 7% of the original edges preserved. In each graph, we show the probability density of the time at which a particular node becomes infected, conditioned on the event that it becomes infected during the epidemic. We show this distribution for two representative nodes (top and bottom) under (A) the localized initial condition and (B) the dispersed initial condition. The top node is in a well-connected part of the network, with typical arrival times ranging from 0.5 to 1.8 in the localized initial condition and from 0.05 to 0.25 in the dispersed initial condition. The bottom node is in a sparser region and more remote from the initial infection, giving it arrival times of 3–8 and 0.2–1.5 in the localized and dispersed initial conditions respectively. All three edge-sampling methods do fairly well at reproducing the shape of these arrival time distributions.

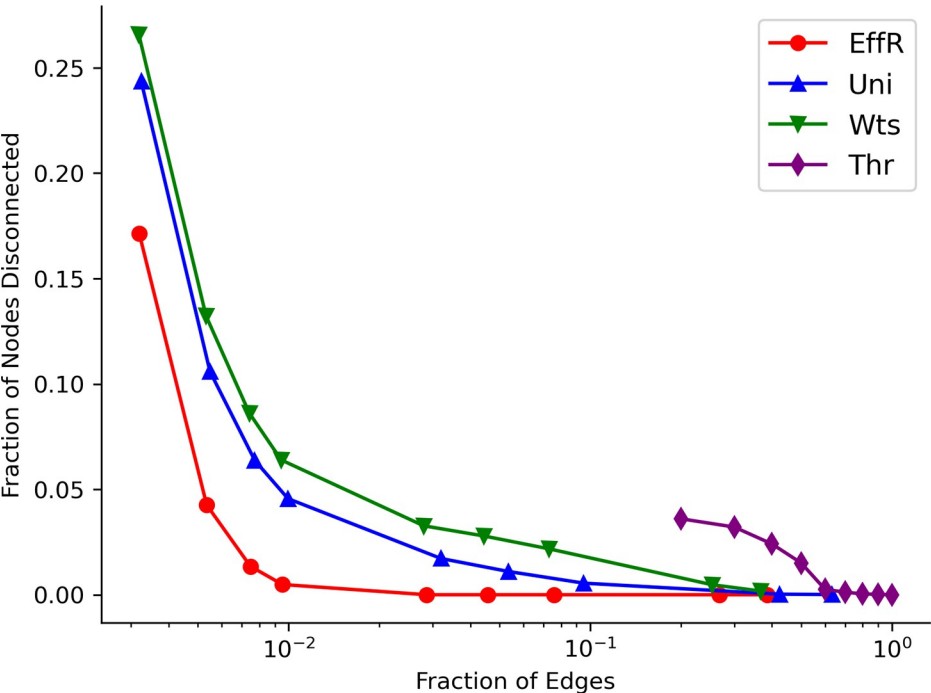

**Fig 5. Sparsification of the U.S. mobility network: average fraction of disconnected nodes.** The average fraction of nodes disconnected from the largest connected component of the network by sparsifiers of different types. The horizontal axis shows the fraction of edges of the original network preserved by the sparsifier. Even when only 1% of the edges are preserved, effective resistance keeps almost all the nodes connected, while uniform and weight-based sampling disconnects 5% and 7% of the nodes respectively.

which classify tracts based on population density, level of urbanization, and daily commuting. As shown in Fig 6, sampling by effective resistances performs well in all 10 types, indicating that it is faithful to a wide range of network structures and mobility patterns. Specifically, sampling by effective resistances achieves the minimum Arrival Time Error Score across all RUCA categories in both the core and periphery of dense metropolitan-, micropolitan-, or town-like areas. As we discuss in the next section, the other sparsification techniques do more poorly in low-commuting areas.

## Discussion

As discussed above, naive thresholding ignores the fact that low-weight edges, individually or in the aggregate, can contribute to disease spread. Removing all such edges can separate communities, isolate rural areas, and remove important long-range links. In contrast, the edge-sampling methods studied here choose at least some low-weight edges, and can thus preserve their contribution to the network structure. Edge-sampling methods also give a principled way to reweight edges to compensate for sparsification, and thus preserve the rates and timescales of the SIR model, thus providing a convenient single parameter slider (the number of samples taken) from a sparse network to the original network in expectation.The question remains why effective resistance is better at preserving infection probabilities and arrival time distributions than weight-based or uniform sampling. We believe this is because effective resistance gives the highest priority to edges where few alternate paths exist—that is, edges that are the only efficient way to travel between their endpoints.

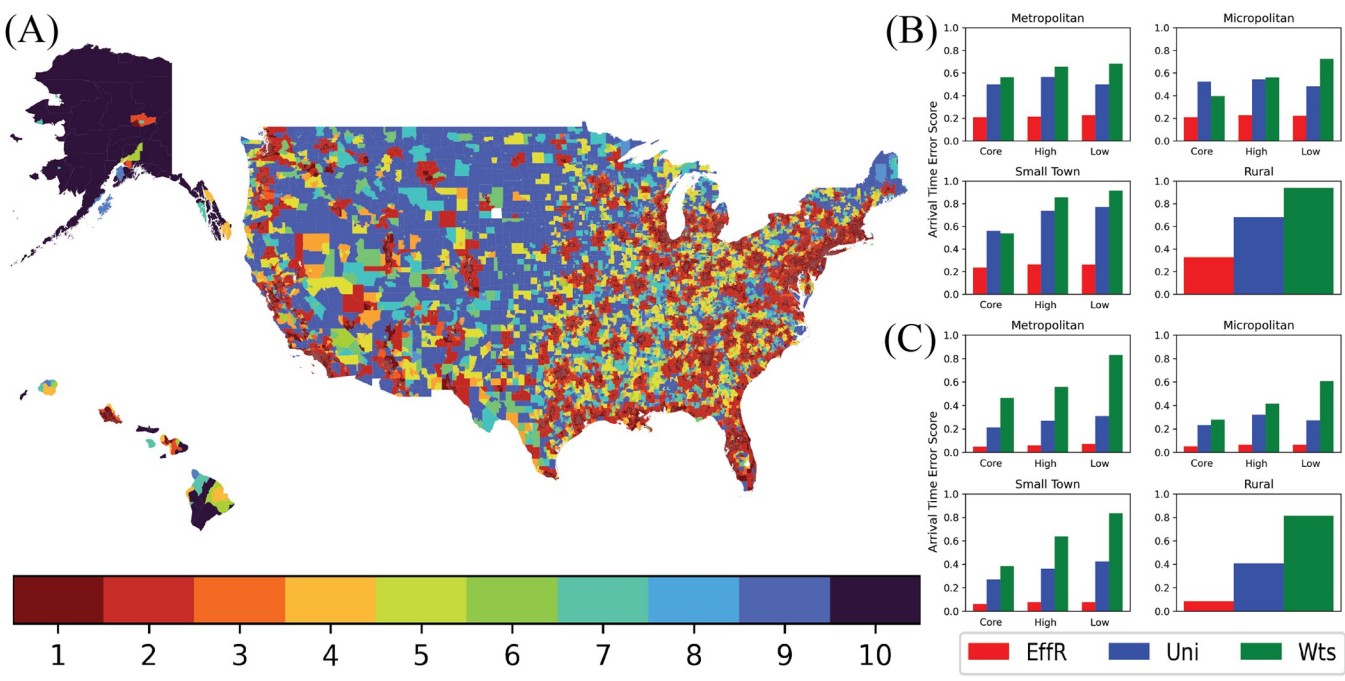

**Fig 6. Map of U.S. Census Tracts by RUCA category.** (A) map of the U.S. where each census tract is color-coded according to its RUCA designation: 1, 2, 3 are metropolitan core, high, and low commuting, respectively, 4, 5, 6 are micropolitan core, high, and low commuting, respectively, 7, 8, 9 are small town core, high, and low commuting, respectively, and 10 is rural. On the right, we show the arrival time error for sparsifiers that sample $s = 0.1m$ edges across the 10 RUCA categories for (B) the localized initial condition and (C) the dispersed initial condition. Across all RUCA categories, effective resistance performs better than uniform or weight-based sampling. This effect is especially pronounced for low-commuting areas, which have fewer or lower-weighted edges connecting them to the rest of the network. Base map provided by U.S. Census Bureau (https://www2.census.gov/geo/tiger/TIGER2016/TRACT/) and is covered under public domain.

For instance, consider a sparse, tree-like region of the network, with few short loops. For each edge we have $R_e \approx 1/w_e$ so the product $w_e R_e$ is close to its maximum value of 1. Thus, effective resistance recognizes the essential role each edge plays in the structure of this region and it gives these edges a priority at least as high as any others in the network. In contrast, weight-based and uniform sampling may let this region become disconnected, dropping nodes with few neighbors or low weighted degree from the network backbone. They will also tend to under-sample edges in such regions, focusing on other regions with larger total edge weight or simply more edges. Similarly, effective resistance recognizes the importance of long-range edges that are the only easy way to cross from one part of the world to another, since we again have $w_e R_e \approx 1$. Weight-based sampling gives these edges low priority if $w_e$ is small, and uniform sampling will keep them only by chance, despite their importance in spreading disease at large scales across the network.

Now consider a dense region of the network with many short loops, and thus many competing paths of various weights between most pairs of nodes. In such a region we need to preserve edges that represent the most-likely paths for disease spread. Uniform sampling ignores both topology and edge weights, so even if it keeps the region connected it makes no attempt to sample more important edges. Weight-based sampling is a reasonable heuristic, since high weight means a high probability of transmission, but, unlike effective resistance, it does not take alternate paths into account and may disconnect nodes that only have edges with low weight.

This picture is borne out by the performance of these sparsification methods in different census tract types (Fig 6). In low-commuting areas, whose network structure is sparser (according to topological degree) and/or lower weight (according to weighted degree),

uniform and weight-based sampling disconnect a larger fraction of nodes than effective resistance. This results in the Arrival Time Error Score of nodes with low infection probability being higher in weight-based sampling than either uniform sampling or sampling by effective resistances. In high-commuting areas, even though it keeps the network more connected, uniform sampling has a higher error than effective resistance in the arrival time distribution conditioned on a nonzero infection probability, indicating that it is not choosing the most important edges for disease spread.

Fig 6 also shows that the distinction between core, high-, and low-commuting areas is more important than the distinction between metropolitan, micropolitan, small town, or rural. Thus, regardless of the level of urbanization, effective resistance does a better job of preserving network structure and epidemic dynamics than the other methods. It keeps nodes connected even if they are in sparser, lower-weight regions, and it selects edges with high structural importance even if they have low weight.

We note that it is common in the literature, including in more sophisticated thresholding methods, to require that every node keeps at least one edge in an effort to keep the network connected (e.g., [15]). One could add that requirement to any of these methods, but it is unclear how to reweight these edges of last resort in a principled manner.

## Conclusion

At a practical level, sparsification significantly reduces computation time. Our simulation takes about 12 minutes for a single run on the original network, and only about 1.75 minutes on a sparse network with about 7% of the original edges. While other implementations may be faster overall, a similar speedup will apply.

Epidemic simulations typically take a total of $O((n + m) \log n)$ time for networks with $n$ nodes and $m$ edges (e.g., [5,46]. This is because there are $n + m$ possible recovery and infection events, and they can be managed with a data structure where events can be added or retrieved in $O(\log n)$ time. For dense networks where $m = O(n^2)$, the typical running time is then $O(m \log n)$, or linear in $m$. Thus, sparsification can be used as a preprocessing step for a wide range of epidemic models, reducing their computation time by roughly the same ratio as the fraction of edges preserved. We note a recent SIR algorithm [47] that takes $O(m \log \log n)$ time using more sophisticated data structures. This algorithm would also benefit from sparsification since all $m$ edges have to be insert into these data structures at the outset.

Beyond this practical application, these results shed light on which edges are the most important for disease spread. In particular, they suggest that effective resistance is a better guide to an edge's importance than its weight in the epidemic context. We find it interesting that a technique designed to preserve linear-algebraic properties of a weighted graph also preserves nonlinear stochastic dynamics of an epidemic model. More generally, effective resistance belongs to a rich class of sparsifiers which seek to preserve dynamical, rather than topological, properties of a graph.

One caveat is that, while this method of sparsification preserves epidemic dynamics, it can obscure the original edge weights. For instance, if there is a bundle of low-weight edges that cross between two communities, the Spielman–Srivastava algorithm will choose one of them and give it a high weight, in essence designating it as the representative of the entire bundle. This makes sense in contexts like epidemic spreading where bundles of parallel edges can work together and act as one high-weight edge. But in some other contexts, such as genetic regulatory networks, where the goal is to understand the functional role of each link and where edge weights are of independent scientific interest, weight-preserving methods like those of [16,17] may be more appropriate.

We believe further work in this area, in both epidemiology and other biological network models, will help us understand how to identify which edges are important to a given dynamical process.

## Acknowledgments

This work was carried out as part of an REU (Research Experience for Undergraduates) program at the Santa Fe Institute under the mentorship of Cristopher Moore and Maria Riolo

## Author Contributions

**Conceptualization:** Alexander Mercier, Cristopher Moore.

**Data curation:** Alexander Mercier.

**Formal analysis:** Alexander Mercier, Samuel Scarpino, Cristopher Moore.

**Funding acquisition:** Cristopher Moore.

**Investigation:** Alexander Mercier.

**Methodology:** Alexander Mercier, Samuel Scarpino, Cristopher Moore.

**Project administration:** Alexander Mercier, Samuel Scarpino, Cristopher Moore.

**Resources:** Alexander Mercier, Samuel Scarpino.

**Software:** Alexander Mercier.

**Supervision:** Samuel Scarpino, Cristopher Moore.

**Validation:** Alexander Mercier, Cristopher Moore.

**Visualization:** Alexander Mercier.

**Writing – original draft:** Alexander Mercier, Samuel Scarpino, Cristopher Moore.

**Writing – review & editing:** Alexander Mercier, Samuel Scarpino, Cristopher Moore.

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
