## [Decision Letter · Decision Letter 0]

10 Jun 2022

Dear Mr. Mercier,

Thank you very much for submitting your manuscript "Effective resistance against pandemics: Mobility network sparsification for high-fidelity epidemic simulations" for consideration at PLOS Computational Biology. As with all papers reviewed by the journal, your manuscript was reviewed by members of the editorial board and by several independent reviewers. The reviewers appreciated the attention to an important topic. Based on the reviews, we are likely to accept this manuscript for publication, providing that you modify the manuscript according to the review recommendations.

Sincerely,

Feng Fu

Associate Editor

PLOS Computational Biology

Virginia Pitzer

Deputy Editor-in-Chief

PLOS Computational Biology

[LINK]

Reviewer's Responses to Questions

**Comments to the Authors:**

Reviewer #1: In this manuscript, the authors discuss a network sparsification method implemented in the context of epidemiology. The method is built upon the earlier algorithm from Spielman and Srivastava, which uses effective resistance to sample the edges. In this paper, the authors simulated the SIR model on a real-world mobility network. They showed that the effective resistance sampling method could preserve the behavior of the SIR model. In addition, the authors examined other simpler sparsification algorithms and found that using effective resistance provides the most accurate simulations. Network sparsification is an interesting and important topic, especially when simulating large-scale epidemical models. Thus, the findings within this paper are potential of great practical importance.

Detailed comments:

1. In the introduction, the authors stated, “It outperforms the simpler edge-sampling methods based on uniform probabilities and edge weights, as well as the naïve thresholding approach.” Simply saying “outperform” is a little vague. I think it would be better if the authors could explain precisely how the effective resistance sampling method is better than the other methods. It seems that the effective resistance sampling method is more accurate to capture the behavior of the SIR model, but it is more computationally expensive than the other simpler sparsification algorithms.

2. Figure 1: Is the color coded on the same scale for the figures on the left and right? The edges look in the same darker blue in the figure on the left.

3. It would be better if the authors could provide some details of the network in the caption of Figure 1. For example, each node represents a census tract, define q as the fraction of edges wanted to preserve, etc.

4. Figure 2: There are blue and black dots in the figure. What does the color represent? Moreover, weight-based sampling method performed almost as good as the effective-resistance method, at least when measured by R-squared. I think the authors should mention this observation in the text.

5. The authors chose the same q values for the three edge-sampling methods and simulated the methods for each value of q. So in Figures 3 and 5, if I draw a vertical line on each chosen value of q, should three different dots lie on the vertical line? It seems that the dots are aligned for q values below 0.01 but not for the values above 0.01.

6. Figure 3: In the caption, the authors mentioned: “the shaded regions corresponds to one standard deviation of the average.” However, I didn’t see the shaded regions in the figure on my end. Also, “corresponds” should be “correspond”.

7. I don’t understand Figure 4 very easily. Please provide a better description. What does the y-axis represent? How are these two nodes chosen? Is it meaningful that all sampling methods have overlapping curves for one node under the localized initial condition but not the dispersed initial condition?

Reviewer #2: The authors propose a new statistical method for weighted graph sparsification, with a focus on preserving epidemic spread dynamics. The paper is very well written and indeed a pleasure to read. The method is of clear relevance to network science and computational biology as so many of the latter's analytical pipelines depend on large networks that benefit from sparsification---not just mobility or other networks involved in epidemic simulations. As such, I recommend publication with some minor suggested edits.

My main comment relates to a comparison with methods we developed in our group. Thus, for maximum transparency, I sign this review. I could not agree more with the authors about the need for principled network reduction methods that preserve essential network dynamics and the hierarchical structure of networks. Two of the methods we have proposed are cited by the authors: distance backbones [Ref. 16] and the effective graph [Ref. 17]. The most related to this paper is the former methodology. While the effective resistance methods is statistically-principled ("the weighted adjacency matrix and graph Laplacian of \\tilde {G} are equal, in expectation, to those of G"), distance backbones are algebraically-principled (and parameter-free). Indeed, distance backbones are unique for a given distance function (typically in network science we sum distance edges resulting in the unique metric backbone, but other distances are possible), while effective resistance leads to different sparsifications in each run (similarly to the disparity filter proposed by Serrano, Boguna and Vespignani [Ref. 15] but likely more efficient in preserving spreading dynamics given the preservation of essential connectivity), and also changes the original edge weights.

Perhaps more importantly, the distance backbone is guaranteed to preserve the entire distribution of shortest paths intact and edge weights, not only the network connectivity --- even though distance backbones are also typically very small [Ref. 16]. The authors say (page 3) that their "strategy helps keep the network connected and preserves its global structure." So, it would be interesting to understand how the effective resistance sparsification affects the original distribution of shortest paths? Figure 4 tallies the fraction of disconnected nodes, which is related, but not the distribution of shortest paths. This suggests that effective resistance preserves the distance backbone for a large range of the fraction of edges removed (unlike the other methods) as the backbone would also keep all edges connected, but the impact on shortest paths may occur for smaller fractions of edges removed. So, while the effective resistance "sparsifier preserves the linear properties of the original networks in expectation," the distance backbone does not affect any shortest path on the network (nor the original distance weights). This speaks to the synergy between the two concepts, and of course I do not expect the authors to run simulations to compare the two methods in this paper. However, the impact of effective resistance sparsification on the distribution of shortest paths is a reasonable question to consider---in addition to the uniqueness of the distance backbone.

Another related question is what to do with the results of the effective resistance sparsifier since it changes the original weights? In the case of the epidemic models this is (more or less) clear because we tend to be more interested in the dynamical observables reported (time to infection, infection probability, etc.) But what about cases where the original weights are meaningful? For instance, in brain connectome or gene regulation networks (as in [Ref. 16]) the original edge weights have specific experimental significance. The distance backbone preserves the edge weights and thus their experimental significance. The authors could discuss or suggest how that would be handled with their sparsifier.

The authors say, in regards to Refs 16&17, that they "know of no rigorous results on whether these techniques preserve dynamical behavior." The effective graph methodology [Ref. 17] is also principled (logically-principled based on the Quine-McCluskey algorithm) , but it applies to networks with node dynamics (such as automata networks used in systems biology). The effective graph is unique because our measure of effective connectivity is a parameter (not a statistic) of the dynamics of Boolean functions. Preserving dynamical behavior is the whole point of Ref. 17, indeed of the whole approach. Because the networks analyzed in this paper have no node-dynamics and are rather used to study spreading dynamics (dynamics on networks rather than dynamics of networks), the effective graph is not directly comparable to the effective resistance methodology (albeit the similar name). Still, one cannot say the latter was not studied in regards to preserving dynamical behavior (our recent discussion in https://doi.org/10.1093/bioinformatics/btac360 could be relevant here).

As for weather the distance backbone methodology [Ref. 16] was studied to preserve dynamical behavior (in the sense of dynamics on networks) is a curious thing since https://doi.org/10.1101/2022.02.02.478784 is under review in this same journal, and of course the authors would not know about it. Still, the utility of the distance backbone in epidemic models has also been studied. I would venture that the effective resistance sparsifier assumes propagation by a particular distance (a resistance distance) which makes a lot of sense for epidemic spread, whereas the distance backbone methodology at large can consider any distance (our under review work on epidemic spread considers only the metric distance). I posit that it should be possible to derive a "resistance backbone" that is unique (not sampled) , but that is clearly an idea for future work---again, supporting the synergy and complementarity between the two methods, which in my view are both very relevant and useful.

Minor comments:

I would welcome a little more justification for the values of the edge-sampling proportion q used. In particular, why only 0.1, 0.55, and 1 for anything above 10%? Probably because there is not much difference between methods tested (safe thresholding) in that range? By the way, since other figures are derived for q=0.1 which led to near 7% edges preserved (right?), a figure with q vs actual % od edges that remain would be useful even if in supporting materials as I imagine this may different from network to network depending on how many (essential) bridges they have (and other properties).

I don't quite get what all the four panels are in Figure 4. A and B sides are explained, but not upper and lower panels.

Really liked the use of the Wasserstein distance for comparisons.

Luis M. Rocha

**Have the authors made all data and (if applicable) computational code underlying the findings in their manuscript fully available?**

Reviewer #1: Yes

Reviewer #2: Yes

PLOS authors have the option to publish the peer review history of their article (what does this mean?). If published, this will include your full peer review and any attached files.

Reviewer #1: No

Reviewer #2: **Yes: **Luis M. Rocha

Figure Files:

Data Requirements:

Reproducibility:

References:

---

## [Decision Letter · Decision Letter 1]

12 Oct 2022

Dear Mr. Mercier,

We are pleased to inform you that your manuscript 'Effective resistance against pandemics: Mobility network sparsification for high-fidelity epidemic simulations' has been provisionally accepted for publication in PLOS Computational Biology.

Best regards,

Feng Fu

Academic Editor

PLOS Computational Biology

Virginia Pitzer

Section Editor

PLOS Computational Biology

Reviewer's Responses to Questions

**Comments to the Authors:**

Reviewer #1: Thank authors for the hard work. It's a pleasure reading this manuscript. All my previous comments have been addressed.

Reviewer #2: Thank you so much for addressing all my comments. I am fully satisfied with the responses and excited for the possibilities this research (and synergy with our own methods) brings to network science and the study of the dynamics of epidemic spread on networks.

**Have the authors made all data and (if applicable) computational code underlying the findings in their manuscript fully available?**

Reviewer #1: Yes

Reviewer #2: Yes

PLOS authors have the option to publish the peer review history of their article (what does this mean?). If published, this will include your full peer review and any attached files.

Reviewer #1: No

Reviewer #2: **Yes: **Luis M. Rocha

---

## [Editor Report · Acceptance letter]

28 Oct 2022

PCOMPBIOL-D-22-00323R1 

Effective resistance against pandemics: Mobility network sparsification for high-fidelity epidemic simulations

Dear Dr Mercier,

I am pleased to inform you that your manuscript has been formally accepted for publication in PLOS Computational Biology. Your manuscript is now with our production department and you will be notified of the publication date in due course.

With kind regards,

Zsofia Freund
